# Device-Measured Sedentary Behavior, Physical Activity and Aerobic Fitness Are Independent Correlates of Cognitive Performance in Healthy Middle-Aged Adults—Results from the SCAPIS Pilot Study

**DOI:** 10.3390/ijerph16245136

**Published:** 2019-12-16

**Authors:** Maria M. Ekblom, Örjan B. Ekblom, Mats Börjesson, Göran Bergström, Christina Jern, Anders Wallin

**Affiliations:** 1The Swedish School of Sport and Health Sciences, 11486 Stockholm, Sweden; 2Department of Neuroscience, Karolinska Institutet, 17177 Stockholm, Sweden; orjan.ekblom@gih.se; 3Department of Neuroscience and Physiology, the Sahlgrenska Academy, University of Gothenburg, 40530 Gothenburgh, Sweden; mats.brjesson@telia.com (M.B.); anders.wallin@neuro.gu.se (A.W.); 4Center for Health and Performance, Department of Food, Nutrition and Sports Science, 40530 Gothenburgh, Sweden; 5Sahlgrenska University Hospital/Ostra, University of Gothenburg, 41345 Gothenburgh, Sweden; 6Department of Molecular and Clinical Medicine, University of Gothenburg, 40530 Gothenburgh, Sweden; goran.bergstrom@hjl.gu.se; 7Department of Clinical Physiology, Sahlgrenska University Hospital, 41345 Gothenburgh, Sweden; 8Department of Laboratory Medicine, Institute of Biomedicine, the Sahlgrenska Academy, University of Gothenburg, 41345 Gothenburg, Sweden; christina.jern@neuro.gu.se; 9Department of Clinical genetics and genomics, Sahlgrenska University Hospital, 41345 Gothenburgh, Sweden; 10The Memory Clinic, Sahlgrenska University Hospital, 40530 Gothenburgh, Sweden

**Keywords:** cognitive functions, physical activity, sedentary behavior, exercise, accelerometry

## Abstract

High aerobic fitness, more moderate to vigorous physical activity (MVPA) and less sedentary behavior (SED) have all been suggested to promote cognitive functions, but it is unclear whether they are independent predictors of specific cognitive domains. This study aimed to investigate to what extent aerobic fitness MVPA and SED are independently associated with cognitive performance among middle-aged Swedish adults. We acquired device-based measures of aerobic fitness, cognitive performance and percent daily time spent in MVPA and SED in Swedish adults (*n* = 216; 54–66 years old). Aerobic fitness was associated with better performance at one out of two tests of speed/attention and one out of four tests of executive attention, and with worse performance at one of seven tests of memory. Increasing %MVPA was associated with better performance at one out of seven tests of memory and two out of three tests of verbal ability, whereas increasing %SED was associated with better performance at all four tests of executive attention and four out of seven tests of memory. These findings suggest that aerobic fitness, %MVPA and %SED are partly independent correlates of cognitive performance. To fully understand the association between SED and performance at several tests of cognitive function, future investigations might attempt to investigate intellectually engaging SED (such as reading books) separately from mentally undemanding SED (such as watching TV).

## 1. Introduction

Individuals with higher aerobic fitness have been shown to present with better academic performance [1], higher cognitive function [2,3,4], lower risks for early onset dementia [5], mild cognitive impairment [5] and stroke [6]. While animal studies generally show robust effects of exercise interventions, the effect sizes in human intervention studies have been more modest, with executive functions being the most responsive to aerobic exercise [7].

Although it is typically assumed that physical activity mainly promotes cognitive functions via its effect on aerobic fitness [8], which in turn improves cerebrovascular function [2], other mechanisms have also been suggested [9,10]. For example, both higher fit and more physically active individuals show greater hippocampal, prefrontal cortex and basal ganglia volume [11,12] than less fit and inactive controls, which together with extensive evidence from animal studies suggests that physical activity triggers the release and action of neurotrophic growth factors that promote neural tissues of importance for cognitive functions [10].

Aerobic fitness is partly genetically determined [13], although the magnitude of the genetic impact has been disputed [14]. However, studies report a high variability in habitual physical activity levels among individuals with the same fitness level [15]. There is also a large variability in sedentary behaviors (SED), such as sitting or lying among individuals with similar levels of physical activity [16]. There is now evidence to suggest that aerobic fitness, physical activity and SED are interdependent correlates of many health outcomes, but there is a paucity of studies investigating the relative importance of physical activity behavior (such as moderate to vigorous physical activity (MVPA) or SED and aerobic fitness for promoting brain health and cognitive function). Gill et al. (2015) [2] recently suggested that the association between aerobic fitness and cognitive function is partly mediated via (improved) cerebrovascular function (such as better cortical circulation, cerebral vascular-motor reactivity, less atherosclerosis, better endothelial functions etc.), whereas the association between self-reported, retrospective lifetime physical activity and cognitive function is not. Although this is very interesting, a major limitation to that study was the subjective assessment of physical activity.

SED has been defined as any awake behavior spent in sitting or lying (at <1.5 metabolic equivalents) [17]. A recent review summarized that self-reported SED is associated with worse cognitive functions [18], while a recent investigation using device-based measures of SED was not able to replicate such associations [19]. These findings underline the importance of assessment methods. Some preliminary evidence suggests that having more non screen SED is directly associated with a higher grade point average, but at the same time indirectly related to lower grade point average, with the latter being mediated via aerobic capacity [20].

Increased knowledge on the relative importance of aerobic fitness and device-measured physical activity behavior, including sedentary time, for cognitive function, is important, and may aid PA-implementation and intervention.

The aim of the current study was to investigate to what extent aerobic fitness and device-based measures of daily time spent in MVPA and SED are independently associated with performance at various cognitive tests, controlling for age, education, tobacco smoking status and being born in or outside of Sweden. We hypothesized that aerobic fitness would primarily be associated with the cognitive tests, where performance relies on processing speed and executive attention, and that %SED and %MVPA, while in the long run influencing aerobic fitness, would also be independent correlates of cognitive performance.

## 2. Materials and Methods

### 2.1. Participants

This study was performed on a subpopulation of the Swedish CArdioPulmonary bioImage Study (SCAPIS) pilot study, which was conducted at the Sahlgrenska University Hospital in Gothenburg, Sweden in 2012 [21]. Men and women participating in the SCAPIS pilot study (*n* = 1089) were invited to participate in one additional test session of cognitive function. Out of these, 326 performed all cognitive tests.

A total of 85 were excluded because they did not have fitness data, and a further 25 because they did not have accelerometry data. In total, 96 men and 120 women were included in the analyses. The study was approved by the Regional Ethics Committee in Gothenburg (Approval numbers 638-16 and 734-13) and was carried out in accordance with the Declaration of Helsinki of 1975 and 1983.

### 2.2. Questionaire

As part of the SCAPIS pilot study, all participants filled in a detailed questionnaire [21]. For the purpose of this study, we used the questions on education, current smoking status and information on whether they were born in or outside of Sweden, in addition to simple, basic data (age and gender).

For education, a self-reported number of years of education was used. Self-reported active smoking was dichotomized into smokers or non-smokers, where non-smokers included former smokers. Participants also filled out where they were born. Using this information, they were then classified as “Born in Sweden” or “Born outside of Sweden”.

### 2.3. Cognitive Test Battery

A battery of cognitive tests designed to adequately investigate the domains simple speed/attention, executive attention, learning/memory and verbal ability, was used. The tests used for these domains are specified under each heading below.

#### 2.3.1. Simple Speed/Attention (*n* = 2)

The tests tapping simple speed/attention were Trail Making Test (TMT) A, consisting of a timed pencil-tracing of a numbered path (Partington and Leiter, 1949, from [22]) and Stroop Test Victoria version, part 1, consisting of a timed verbal naming of 24 colored dots.

#### 2.3.2. Executive Attention (*n* = 4)

The tests tapping executive attention were Stroop Test Victoria version part 2, consisting of a timed naming of words’ print color, part 3, consisting of naming of the print color of color words [22]; TMT B, consisting of a timed pencil-tracing of numbers and letters, 1-A-2-B, etc. [22]; and a Symbol Digit Modalities Test (SDMT), pencil in numbers in rows of blank squares, guided by nine corresponding symbols (Cognitive Assessment Battery (CAB) version [23]), same symbols but different numbers cf. the original [22]).

#### 2.3.3. Learning/Memory (*n* = 7)

The tests tapping learning/memory were short story memory test, CAB [23] but now revised (CAB-R) to score both per verbatim and synonym, for both immediate and delayed recall; and Rey Complex Figure (RCF) immediate, delayed recall, and recognition [22].

#### 2.3.4. Verbal Ability (*n* = 3)

The tests tapping verbal ability was a Controlled Oral Word Association Test, letters F-A-S, which consisted of naming as many words as you can for a given letter in one minute (COWAT-FAS, [22]) Category Fluency Test (CFT); Animal Naming, consisting of verbally naming as many animals as you can in one minute [22]; and a modified 30 item Boston Naming Test [23], featuring redrawn images from the original Boston Naming Test [24].

All tests were administered in Swedish under the supervision of a licensed psychologist, by a psychologist in training, a trained researcher, or a health coach. All scores were re-checked by a licensed psychologist. Patients used their normal glasses and/or hearing aids. No formal assessment of hearing or eyesight was performed. One participant reported impaired color perception, but as the tests depending on color (Stroop) indicated a normal result, data from the participant was not excluded from the analysis. Insufficient knowledge of Swedish was the sole exclusion criteria.

All patients presenting a cognitive complaint were offered further medical attention, should they so desire.

### 2.4. Aerobic Fitness

All participants in the SCAPIS pilot study were invited to undertake a submaximal ergometer aerobic fitness test. After a period of rest, participants were fitted with a heart rate monitor (Polar, Kempele, Finland), seated on an individually-adjusted bicycle ergometer (Monark model 828E, Varberg, Sweden) and instructed to pedal at an intensity resulting in a sub-maximal heart rate at a cadence of 60 rpm. Aerobic fitness (VO2max, expressed as mL·min^−1^·kg^−1^) was estimated using the Ekblom Bak method [25,26], which is based on a linear relationship between change in heart rate in response to a change in work rate. Participants with a diagnosed heart condition or taking beta-adrenergic blockers constituted a majority of excluded participants. Other reasons for non-participation in fitness testing included pain (hips, back, knees), obesity and perceived inability to perform the test. For analyses, participants were divided into five equally large groups (pentiles) based on fitness values.

### 2.5. Physical Activity and Sedentary Behavior

Physical Activity (PA) data was collected using an accelerometer (ActiGraph model GT3X/GT3X+, Actigraph LCC, Pensacola, FL, USA) carried in an elastic belt over the right hip. Participants were instructed to wear the accelerometer during waking hours for seven consecutive days from the first study visit. After the seven days, the accelerometer was returned to the laboratory by prepaid mail. The accelerometer was initialized and downloaded using ActiLife v.6.10.1 software (Actigraph LCC, Pensacola, FL, USA). Raw data sampling frequency was set to 30 Hz, and triaxial data were extracted in 60-s epochs with a low frequency extension filter. The accelerometers measure acceleration with the arbitrary unit counts per minute (cpm). Non-wear time was defined as 0 cpm for more than 60 min, while allowing a maximum of 2 min between 0 and 200. Wear time was defined as the non-wear time subtracted from 24 h. The wear time was then further analyzed, defining epochs with cpm < 200 as being spent in SED [27], epochs between 200 and 2089 cpm as low intensity physical activity (LIPA), and epochs > 2089 cpm as MVPA [28]. Average percent time per day spent in intensity-specific categories (%SED, %LIPA and %MVPA), was calculated over the entire studied period. For use in the regression analysis, %MVPA and %SED were further categorized into pentiles.

### 2.6. Statistics

Data were tested for normality using the Shapiro-Wilk test. Non-normal data was transformed to assume normality, using log (Lg) or square root (sqr). Linear regression modeling was then performed to investigate the associations between each cognitive function variable and aerobic fitness, %SED and %MVPA, controlling for age, gender, smoking, education and being born in or outside of Sweden. We chose not to adjust for the components of the metabolic syndrome, as these factors are associated with both aerobic fitness and PA-patterns [29] and thus, are not independent regarding effects on cognition. The strength of the associations were described using the standardized regression coefficients (*β*). Sensitivity analyses were performed to ensure that the direction of the associations was not changed when only participants reporting being born in Sweden were included in the analyses.

## 3. Results

The sample used for analyses consisted of 216 individuals, (120 women and 96 males, aged 54–66 years), having data on all aerobic fitness and physical activity variables and participating in the well-validated neuropsychological tests, assessing a broad spectrum of cognitive functions. There were no significant differences between men and women in age, education, physical activity measures, smoking habits or the proportion of participants born outside of Sweden, while women were less fit and spent less time in SED and vigorous physical activity, and men were more likely to have high levels of triglycerides (Table 1).

Among the tests tapping speed and attention, %MVPA and %SED were not related to any of the outcomes, whereas a higher aerobic fitness tended to be associated with better performance at Stroop 1, but not TMT A (Table 2). It is of note that individuals born in Sweden (*n* = 165) had higher performance at both tests (Table 2). We therefore performed sensitivity analyses using this subset of the study population. In these analyses, the direction of the associations remained. Still, the association between aerobic fitness and Stroop 1 was attenuated (*β* = −0.099, *p* = 0.287) and the association between %SED and TMT A strengthened (*β* = −0.189, *p* < 0.05).

Among the tests tapping executive attention, higher %SED was related to better performance at all tests, while %MVPA was not related to any of the tests, and higher aerobic fitness was related to better performance at Stroop 3 (Table 3). Sensitivity analyses showed that when using only the part of the sample born in Sweden, the direction of these associations remained. Still, in this subsample, aerobic performance also tended to be related to better performance at Stroop 2 (*β* = −0.159, *p* = 0.079) and the association between %SED and Stroop 3 was attenuated (*β* = −0.131, *p* = 0.123).

Among the tests tapping learning and memory, higher %SED was related to better performance in four verbal memory tasks, but was not related to the three visuospatial memory tests (Table 4). Higher %MVPA was related to better performance only in RCF recognition (Table 4), whereas better aerobic fitness was related to worse performance at RCF recognition. Sensitivity analyses showed that when using only the part of the sample born in Sweden, the direction of these associations remained. %SED was an even stronger correlate of memory in this subsample with the association between %SED and delayed verbatim memory now reaching significance (*β* = 0.210, *p* < 0.05).

Among the tests tapping verbal ability, more %MVPA was related to better abilities at two of the three tests (Table 5), while neither %SED nor aerobic fitness was significantly associated with any of these tests. Sensitivity analyses showed that when using only the part of the sample born in Sweden, the direction of these associations remained. Still, in this subsample more %SED tended to be associated with better performance at animal naming (*β* = 0.141, *p* = 0.098) and category naming (*β* = 0.144, *p* = 0.086), the association between %MVPA and category naming was attenuated (*β* = 0.128, *p* = 0.126) and a tendency for better aerobic capacity to be related to better performance at BNT appeared (*β* = −0.171, *p* = 0.054).

In summary, higher aerobic fitness was associated with better performance at some, but not all, tests of speed, attention and divided (executive) attention, but to worse performance at RCF recognition. High %MVPA was related to better RCF recognition and better verbal ability, while %SED was not associated with speed, but more %SED was related to better performance at all tests of divided (executive) attention and all verbal, but not the visuospatial, memory tests. Although being born in Sweden was a strong correlate of better performance at most cognitive tests, sensitivity analyses showed that when using only the part of the sample born in Sweden, the direction of associations remained the same between on the one hand %MVPA, %SED and aerobic fitness, and the cognitive variables on the other.

## 4. Discussion

The main findings of the current study were that aerobic fitness, %MVPA and %SED were associated with different cognitive domains, in unselected middle-aged subjects. In line with previous investigations in primarily elderly populations [30], aerobic fitness was associated with better performance at some of the tests of speed and executive attention, while controlling for age, education, smoking, being born in or outside of Sweden and device-measured %MVPA and %SED. When controlling for the same variables, higher %MVPA was not associated with any of the tests of executive attention, but was instead associated with better verbal ability. Interestingly, more %SED (i.e., sitting more) was associated with higher scores at all but one test of executive attention and at all verbal memory tests.

### 4.1. Why Ivestigate Independent Association?

Rather often, aerobic fitness is used as a proxy for physical activity and vice versa, physical activity as a proxy of aerobic fitness. In the same way, meeting the physical activity recommendations is often interpreted as being equivalent to not having a sedentary lifestyle. This is unfortunate, since the same individual can have both high %SED and high %MVPA [31]. Further, there is strong empirical evidence in favor of a genetic component of aerobic fitness.

As a consequence of this, there is a large variation in fitness between individuals with similar levels of MVPA and/or SED and reversely there are large variations in habitual physical activity and sedentary behaviors among individuals with the same fitness level. The current study attempted to enhance the understanding of how physical activity patterns and aerobic fitness, in more detail, are associated with cognitive function. Independent associations of each of these variables to individual cognitive tests were identified. The associations differed across cognitive outcome variables, sometimes even within the same cognitive domain. In this study, it is possible that this signifies that some of the cognitive tests are designed to identify difficulties rather than to measure abilities.

### 4.2. Independent Correlates of Speed and Executive Attention

The association between aerobic fitness and high performance at tests of speed and executive attention was expected, since this finding has been described in the literature, for a working age population [4], and in elderly populations [30]. There was however no independent association between %MVPA and executive attention. While the sample size of this study was small, these findings are consistent with the notion that %MVPA does not enhance executive attention independently of aerobic fitness [32]. The independent association between higher %SED and better performance at all tests of executive attention was somewhat unexpected, since %SED has been associated with increased risk for metabolic syndrome [29], which leads to increased vulnerability to vascular disease including atherosclerosis. This association could also mean that a large proportion of the SED was spent in cognitively engaging tasks. That is, while sitting is potentially detrimental for cognition abilities, this could possibly be outweighed by activities stimulating cognition, such as reading. As such, in regards to effects on cognition, the metabolic effects of inactivity seem to be subordinate to the cognitive activity undertaken during the activity. It is unknown if the same relation is true for MVPA. While this study was limited by its sample size, future studies should attempt to investigate whether aerobic fitness or MVPA moderates the association between %SED and cognition. Sandbakk et al. [33] recently showed that self-rated sedentary behavior was associated with cardiovascular disease and mortality, only among individuals with an age specific low fitness level. In line with this Katzmarzyk et al. [34] also showed that self-rated sedentary behavior was a weaker predictor of mortality in physically active as compared to physically inactive individuals.

### 4.3. Independent Correlates of Memory and Learning

Apart from being important for executive attention, higher %SED was also associated with better performance at all verbal memory tests. This appears to be consistent with the idea that intellectual activities of importance for vocabularies, such as reading [35], can make up an influential part of time spent in sedentary behavior. The visuospatial memory and learning tests were only associated with education, except for the RCF recognition, which was positively associated with %MVPA and negatively associated with aerobic fitness. The latter finding gives some support to the notion that %MVPA supports RCF recognition through mechanisms other than by aerobic fitness. Such mechanisms could include enhanced neuroplasticity from increased exercise-induced bioavailability of neurotrophic factors such as brain-derived neurotrophic factor [36].

### 4.4. Independent Correlates of Verbal Ability

While not of importance for performance on tests tapping speed and executive attention, higher %MVPA was instead independently associated with better performance at two out of three tests of verbal ability. This has previously been reported using self-rated estimates of physical activity in a mixed middle age and elderly population [37], but has to our knowledge not been demonstrated in a middle aged population using devise-based measures of physical activity. The tests of verbal ability were, as expected, strongly influenced by whether the individuals were born in or outside of Sweden. The directions of the associations did not change when we considered only the part of the sample born in Sweden, but the strength of the associations did. Care should therefore be taken when interpreting these findings.

### 4.5. Methodological Considerations

The current study has some weaknesses. The sample consisted of more females than males. Gender differences are sometimes found in cognitive functions. A recent meta-analysis of randomized controlled trials (RCTs) concluded that RCTs with a larger proportion of women show larger effect sizes for executive function and global cognition [38]. The cross-sectional nature of this study also makes it impossible to investigate causality. Alongside the idea of a positive effect of physical activity on cognitive function is the idea of the reversed causality by which executive functions are important components in the self-regulation needed to sustain positive lifestyle behaviors. It has been suggested that the association between aerobic fitness and cognition may be reciprocal, and that individuals with better executive functions achieve a better aerobic fitness through a healthier, more active lifestyle [39,40]. Additional data on context, for example type of sedentary activities (reading, screen time, etc.), would have been an asset, to support the proposed importance of this on cognitive functioning.

The current study also has several strengths. The inclusion of a multiple cognitive test for each cognitive domain and the inclusion of both device-based physical activity behavior and aerobic fitness instead of self-reported physical activity are the most important strengths. Another strength is the careful control for confounders, including information on whether the participants were born in Sweden or not [41], which has recently been shown to influence performance at cognitive tests, most likely due to the tests being performed in the non-native language [41]. Being born in Sweden was associated with higher performance at many of the cognitive tests, but our sensitivity analyses showed that when using only the part of the sample born in Sweden, the direction of the associations between %MVPA, %SED and aerobic fitness to performance at the cognitive tests, remained.

### 4.6. Implications

Sedentary time can work both ways regarding effects on cognition. The consistent finding of more %SED as a correlate of better cognitive functions is in contrast with previous investigations using subjective ratings of sedentary behaviors, but in line with some recently published investigations using device-based measures. Watching TV is sometimes used as a proxy for SED. SED as measured by extensive TV watching has previously been reported to be associated with poorer executive performance and processing speed [42]. However, subjective assessments of sedentary behavior in one domain may not be a valid measure of the total amount of daytime SED [43]. SED has been shown to be a risk factor for metabolic health, that is partly independent of aerobic fitness and MVPA [32] (but see also [44,45]). It would therefore be expected that higher %SED would be associated with worse rather than better cognitive functions, especially among low fit individuals. At the same time, engaging in intellectually challenging activities has also been shown to positively predict cognitive function [46], and since intellectual activities are often sedentary, this may counteract effects of metabolic risk factors on cognitive functions. In line with this, Hallgren et al. [47] recently observed that self-rated time spent in “mentally passive” SED was related to increased rates of depression, while “mentally active” SED was related to decreased rates of depression. To fully understand the interactive effects of cognitive engagement and physical activity patterns on cognitive functions, we suggest that future investigations use combinations of device-based measures of SED and subjective classifications of context.

## 5. Conclusions

The findings of the current study strengthen the evidence of low values of aerobic fitness being an independent correlate of lower performance at some, but not all, tests of executive attention in middle-aged adults. Such associations have also been shown in elderly populations [30]. In addition, %MVPA showed an independent association to verbal fluency, and one out of seven measures of memory, while higher %SED was independently associated with better performance at several cognitive tests tapping executive attention and verbal memory.

These findings are only partly in line with previous investigations relying on self-reported estimates of physical activity and sedentary behavior. Lack of exercise and too much sedentary time are both major causes of chronic disease [48], and aerobic fitness is a strong predictor of cardiovascular morbidity and all-cause mortality [49]. Still, the results of the present study emphasize that aerobic fitness and physical activity behaviors are partly independent correlates of cognitive functions. Further investigations are needed to understand these associations in populations of different ages, and using different measures of physical fitness and cognitive functions. Our findings reinforce that care should be taken when interpreting associations between subjectively rated activity patterns and health outcomes. To fully understand the associations observed between %SED and several of the cognitive functions in this study and others, future investigations should attempt to separate effects of cognitively engaged SED (for example reading) from less cognitively engaging SED. Also, future studies should attempt to investigate whether aerobic fitness moderates the association between SED and cognition.

## Figures and Tables

**Table 1 ijerph-16-05136-t001:** Subject characteristics. Displaying descriptive statistics and comparisons of the participating men and women.

Descriptive Variables	Women (*n* = 120)		Men (*n* = (96))		Sex Difference
	Median	Quartile 1–3	Median	Quartile 1-3	
Age (years)	60.0	59.0–63.8	60.0	58.0–64.0	*p* > 0.05
Aerobic fitness (mL min^−1^ kg^−1^)	28.6	24.1–34.2	35.0	31.0–39.2	*p* < 0.001
Sedentary time (%)	49.7	44.5–58.7	54.3	47.9–61.4	*p* < 0.05
Moderate PA (%)	5.2	3.7–6.6	5.0	3.7–7.3	*p* > 0.05
Vigorous PA (%)	0.1	0.0–0.4	0.2	0.0–1.1	*p* < 0.05
Education (years)	13.7	12.0–15.4	13.0	12.0–16.0	*p* > 0.05
Current smokers (%)	11.5		10.0		*p* > 0.05
Born outside of Sweden (%)	19.1		28.1		*p* > 0.05
Hypertension (%)	20.0		19.8		*p* > 0.05
Known diabetes (%)	1.7		2.1		*p* > 0.05
High triglycerides (%)	5.0		13.5		*p* < 0.05

High Triglycerides were defined as above 2 mmol/L. PA = Physical activity.

**Table 2 ijerph-16-05136-t002:** Standardized Beta-coefficients for linear regressions with neuropsychological tests tapping simple speed/attention.

Independent Variables	LgStroop 1 Adj. R^2^ = 0.343	LgTMT A Adj. R^2^ = 0.112
%MVPA	n.s.	n.s.
%SED	n.s.	n.s.
Aerobic fitness	**−0.108** *	n.s.
Age	n.s.	0.135 **
Education	n.s.	n.s.
Smoking	n.s.	n.s.
Sex	−0.276 ***	n.s.
Born in Sweden	−0.506 ***	−0.285 ***

* *p* ≤ 0.1, ** *p* ≤ 0.05, *** *p* ≤ 0.001. Average time spent in moderate to vigorous physical activity (%MVPA) and sedentary behavior (%SED) and aerobic fitness are expressed in pentiles. A higher value is represented as being a current smoker, being female and being born in Sweden. The standardized beta-coefficients of the associations are in bold where *p* ≤ 0.1 for %MVPA, %SED or aerobic fitness. Adj. = Adjusted.

**Table 3 ijerph-16-05136-t003:** Standardized beta-coefficients for linear regressions with neuropsychological tests tapping executive attention.

Independent Variables	SqrStroop 2 Adj. R^2^ = 0.362	LgStroop 3 Adj. R^2^ = 0.225	SDMT Adj. R^2^ = 0.177	LgTMT B Adj. R^2^ = 0.273
%MVPA	n.s.	n.s.	n.s.	n.s.
%SED	**−0.141 ****	**−0.127 ***	**0.126 ***	**−0.113 ***
Aerobic fitness	n.s	**−0.203 ****	n.s.	n.s.
Age	n.s.	n.s.	−0.148 **	0.188 **
Education	n.s.	n.s.	0.161 **	−0.171 **
Smoking	n.s.	n.s.	n.s.	n.s.
Sex	−0.146 **	−0.251 **	0.135 *	n.s.
Born in Sweden	−0.552 ***	−0.357 **	0.301 ***	−0.448 ***

* *p* ≤ 0.1, ** *p* ≤ 0.05, *** *p* ≤ 0.001. Average time spent in moderate to vigorous physical activity (%MVPA) and sedentary behavior (%SED) and aerobic fitness are expressed in pentiles. A higher value is represented as being a current smoker, being female and being born in Sweden. The standardized beta-coefficients of the associations are in bold where *p* ≤ 0.1 for %MVPA, %SED or aerobic fitness. Adj. = Adjusted.

**Table 4 ijerph-16-05136-t004:** Standardized beta-coefficients for linear regressions with neuropsychological tests tapping learning/memory.

Independent Variables	Memory Verbatim Adj. R^2^ = 0.161	Memory Direct Synonym Adj. R^2^ = 0.196	Memory Delayed Verbatim Adj. R^2^ = 0.127	Memory Delayed Synonym Adj. R^2^ = 0.144	RCF Immediate Adj. R^2^ = 0.045	RCF Delayed Adj. R^2^ = 0.038	RCF Recognition Adj. R^2^ = 0.055
%MVPA	n.s.	n.s.	n.s.	n.s.	n.s.	n.s.	**0.143 ****
%SED	**0.136 ****	**0.137 ****	**0.119 ***	**0.134 ***	n.s.	n.s.	n.s.
Aerobic fitness	n.s	n.s.	n.s.	n.s.	n.s.	n.s.	**−0.183 ****
Age	n.s.	n.s.	n.s.	0.132 *	n.s.	n.s.	n.s.
Education	0.310 ***	0.309 ***	0.255 ***	0.281 ***	0.209 **	0.205 **	0.116 *
Smoking	n.s.	n.s.	n.s.	n.s.	n.s.	n.s.	n.s.
Sex	n.s.	n.s.	n.s.	n.s.	n.s.	n.s.	−0.154 **
Born in Sweden	0.201 **	0.266 ***	0.198 **	0.209 **	n.s.	n.s.	0.212 **

* *p* ≤ 0.1, ** *p* ≤ 0.05, *** *p* ≤ 0.001. Average time spent in moderate to vigorous physical activity (%MVPA) and sedentary behavior (%SED) and aerobic fitness are expressed in pentiles. A higher value is represented as being a current smoker, being female and being born in Sweden. The standardized beta-coefficients of the associations are in bold where *p* ≤ 0.1 for %MVPA, %SED or aerobic fitness. RCF = Rey Complex Figure. Adj. = Adjusted.

**Table 5 ijerph-16-05136-t005:** Standardized Beta-coefficients for linear regressions with neuropsychological tests tapping verbal ability.

Independent Variables	Animal Naming Adj. R^2^ = 0.220	Category Fluency FAS Adj. R^2^ = 0.237	Boston Naming Test Adj. R^2^ = 0.480
%MVPA	n.s.	**0.115 ***	**0.114 ****
%SED	n.s.	n.s.	n.s.
Aerobic fitness	n.s.	n.s.	n.s.
Age	n.s.	n.s.	0.180 ***
Education	n.s.	0.224 ***	0.170 **
Smoking	n.s.	n.s.	n.s.
Sex	n.s.	n.s.	n.s.
Born in Sweden	0.419 ***	0.413 ***	0.532 ***

* *p* ≤ 0.1, ** *p* ≤ 0.05, *** *p* ≤ 0.001. Average time spent in moderate to vigorous physical activity (%MVPA) and sedentary behavior (%SED) and aerobic fitness are expressed in pentiles. A higher value is represented as being a current smoker, being female and being born in Sweden. The standardized beta-coefficients of the associations are in bold where *p* ≤ 0.1 for %MVPA, %SED or aerobic fitness.

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
