# Peer review of "Device-Measured Sedentary Behavior, Physical Activity and Aerobic Fitness Are Independent Correlates of Cognitive Performance in Healthy Middle-Aged Adults—Results from the SCAPIS Pilot Study"

_ijerph, 2019, doi:10.3390/ijerph16245136_

Round 1

Reviewer 1 Report

Thank you for your submission. 

Thank you for submitting this article for review.

The title is a little long, although explains the study.

Abstract is clear and well written. 

Introduction is comprehensive and justifies the aim of the research study.

Methods well written and comprehensive. 

Result clear, justified and explained. 

Discussion- clear consideration for relevance and acknowledgement of strengths and limitations. 

minor comments

lines 81-82 "various cognitive tests, controlling for age, education, being born in or outside of Sweden and smoking"  … the and smoking reads a little odd, assume this is smoking tobacco cigarettes.  perhaps education, cigarette smoker status, and being born in or outside of Sweden, would read better.)

line 148, Physical Activity (PA) not activity

Overall a very nice and well written paper. 

Author Response

Thank you for your positive review and comments. We have addressed each of your comments below and made changes to the manuscript where necessary.

Minor comments from reviewer 1.

Comment 1:

"lines 81-82 "various cognitive tests, controlling for age, education, being born in or outside of Sweden and smoking" … the and smoking reads a little odd, assume this is smoking tobacco cigarettes. perhaps education, cigarette smoker status, and being born in or outside of Sweden, would read better.)"

Response: Thank you for your comment. We have now changed the wording as suggested.

Comment 2:

"line 148, Physical Activity (PA) not activity"

Response: Changed accordingly.

Comment 3. Overall a very nice and well written paper.

Response: Thank you for your positive response and for your comments.

Reviewer 2 Report

The study is well designed and presented. The findings appear to be useful for gerontology studies and should be elaborated using different samples and variables to better understand the associations between physical fitness and cognitive abilities. I strongly advise the authors to include related research in the conclusion part of the study.

Author Response

Comments and Suggestions from reviewer 2.

Comment:

"The study is well designed and presented. The findings appear to be useful for gerontology studies and should be elaborated using different samples and variables to better understand the associations between physical fitness and cognitive abilities. I strongly advise the authors to include related research in the conclusion part of the study."

Response: Thank you for constructive comment. We have now added related research in the conclusion and emphasized that these associations need to be investigated in populations of different ages and using different measures of physical fitness and cognitive functions.

Reviewer 3 Report

There are several issues with the manuscript including not specifying the participants as either midlife or older age.  In one part of the paper it is suggested that there has not been research (like this) among 'elderly' persons [4.2,, lines 275-6]. However, the research by cited (Pantzar et al., 2018) did not have participants who were eldely (they were midlife).  those   If my interpretation of this sentence is incorrect, then the authors should provide further evidence to help support their claim of how this research is different for midlife adults.

The tables (1 and 3) presented should be on one page. It is difficult to read them on separate pages.

The strengths of the study (4.5, second paragraph, lines 327, 328) are not thoroughly described. How are the cognitive battery test and devise measures important strengths?

2.3 lines 105-134: needs to be rewritten as complete sentences

Line 106- battery test- should be plural

Line 330- affect should be effect

Line 356- low aerobic fitness is not previously described in the manuscript. It should be described in the introduction and research related to low aerobic fitness (i.e.walking) should be also referenced. 

Author Response

Thank you for your constructive comments. We have adressed each of your comments below and made changes in the manuscript accordingly.

Comments and suggestions from reviewer 3.

Comment 1:

"There are several issues with the manuscript including not specifying the participants as either midlife or older age. In one part of the paper it is suggested that there has not been research (like this) among 'elderly' persons [4.2,, lines 275-6]. However, the research by cited (Pantzar et al., 2018) did not have participants who were eldely (they were midlife). those   If my interpretation of this sentence is incorrect, then the authors should provide further evidence to help support their claim of how this research is different for midlife adults."

Response: Thank you for constructive comment. We have now attempted to better emphasize the age of our sample and how our findings relate to previous findings. Pantzar et al., used a working age (20-65 yrs) population of office workers, while the current study used a mixed middle-aged (55-66 yrs) population. The age group investigated here is still a bit younger than the elderly (65+) age group that is most often investigated.

Comment 2: "The tables (1 and 3) presented should be on one page. It is difficult to read them on separate pages."

Response: Changed accordingly.

Comment 3:

"The strengths of the study (4.5, second paragraph, lines 327, 328) are not thoroughly described. How are the cognitive battery test and devise measures important strengths?"

Response: We have now attempted to clarify this section. The new text reads:

“The inclusion of multiple cognitive test for each cognitive domain and the inclusion of both device-based physical activity behaviour and aerobic fitness instead of self-report of physical activity are the most important strengths.”

Comment 4:

"2.3 lines 105-134: needs to be rewritten as complete sentences"

Response: We have now rewritten this section as complete sentences.

Comment 5:

Line 106- battery test- should be plural

Response: Changed accordingly.

Comment 6:

"Line 330- affect should be effect"

Response: Changed to “influence”.

Comment 7:

"Line 356- low aerobic fitness is not previously described in the manuscript. It should be described in the introduction and research related to low aerobic fitness (i.e.walking) should be also referenced."

Response: Low aerobic fitness refers to the aerobic capacity that we measured using the submaximal fitness test. We have now clarified the sentence slightly to avoid misunderstandings. We have also added references in the conclusion that highlight the importance of different intensities of physical activity and aerobic capacity for chronic disease, cardiovascular morbidity and mortality.